# Prevention and Control of Swine Enteric Coronaviruses in China: A Review of Vaccine Development and Application

**DOI:** 10.3390/vaccines12010011

**Published:** 2023-12-21

**Authors:** Fanzhi Kong, Huilin Jia, Qi Xiao, Liurong Fang, Qiuhong Wang

**Affiliations:** 1College of Animal Science and Veterinary Medicine, Heilongjiang Bayi Agricultural University, No. 5 Xinfeng Road, Sartu District, Daqing 163319, China; kong001@byau.edu.cn (F.K.); huilinjia@byau.edu.cn (H.J.); xiaoqi0401@byau.edu.cn (Q.X.); 2State Key Laboratory of Agricultural Microbiology, College of Veterinary Medicine, Huazhong Agricultural University, Wuhan 430070, China; 3Key Laboratory of Preventive Veterinary Medicine in Hubei Province, Cooperative Innovation Center for Sustainable Pig Production, Wuhan 430070, China; 4Center for Food Animal Health, Department of Animal Sciences, College of Food, Agricultural and Environmental Sciences, The Ohio State University, Wooster, OH 44691, USA; 5Department of Veterinary Preventive Medicine, College of Veterinary Medicine, The Ohio State University, Columbus, OH 43210, USA

**Keywords:** swine enteric coronavirus, vaccine, transmissible gastroenteritis virus, porcine epidemic diarrhea virus, porcine deltacoronavirus, swine acute diarrhea syndrome coronavirus

## Abstract

Swine enteric coronaviruses (SECs) cause significant economic losses to the pig industry in China. Although many commercialized vaccines against transmissible gastroenteritis virus (TGEV) and porcine epidemic diarrhea virus (PEDV) are available, viruses are still widespread. The recent emergence of porcine deltacoronavirus (PDCoV) and swine acute diarrhea syndrome coronavirus (SADS-CoV), for which no vaccines are available, increases the disease burden. In this review, we first introduced the genomic organization and epidemiology of SECs in China. Then, we discussed the current vaccine development and application in China, aiming to provide suggestions for better prevention and control of SECs in China and other countries.

## 1. Introduction

Coronaviruses (CoVs) are single-stranded, positive sense RNA viruses belonging to the *Orthocoronavirinae* subfamily, *Coronaviridae* family within the *Nidovirales* order. Coronaviruses are classified into four genera: *Alphacoronavirus* (α-CoV), *Betacoronavirus* (β-CoV), *Gammacoronavirus* (γ-CoV), and *Deltacoronavirus* (δ-CoV). They cause a variety of respiratory, enteric, and neurological diseases in humans and animals. Among them, α-CoVs and β-CoVs infect mammals, while γ-CoVs and δ-CoVs primarily infect birds with some mammalian spillover [1,2,3,4]. The coronavirus life cycle includes viral attachment and entry, nonstructural protein expression, genomic RNA replication, transcription, structural protein expression, and viral particle assembly and release [5]. Swine enteric coronaviruses (SECs) include transmissible gastroenteritis virus (TGEV), porcine epidemic diarrhea virus (PEDV), porcine deltacoronavirus (PDCoV), and swine acute diarrhea syndrome coronavirus (SADS-CoV, also known as porcine enteric alphacoronavirus (PEAV) and swine enteric alphacoronavirus (SeACoV)) [6,7]. Although domestic pigs are the main host of SECs, TGEV, PEDV, and PDCoV have also been detected in wildlife, such as PEDV and TGEV from wild pigs and PDCoV from Chinese ferret badgers and Asian leopard cats [8,9]. SEC infections result in high mortality in neonatal piglets if they are born from SEC-naïve sows/gilts that do not provide the specific protective maternal antibodies via colostrum and milk [10,11]. Vaccination/immunization of pregnant sows/gilts to induce protective lactogenic immunity is the most effective approach for preventing and controlling SEC mortality in suckling piglets. In China, multiple lineages/clusters of an SEC are co-circulating on farms and constantly evolving via mutating and recombining, generating new variants [4,7,12,13,14]. Therefore, the development of safe and effective vaccines against SECs is urgently needed. In this review, we discuss the strategies for vaccine development and vaccine application status in China.

### 1.1. Genomic Organization of SECs

The genome of an SEC is 25–29 Kb in length with 5′-capped and 3′-polyadenylated untranslated regions (UTRs). Five open reading frames (ORFs) encode two polyproteins (pp1a and pp1ab), which will be protease-processed to generate up to 16 nonstructural proteins for virus replication and transcription, and four structure proteins: spike (S), envelope (E), membrane (M), and nucleocapsid (N) proteins [14,15,16,17]. In addition, accessory genes exist between structural protein genes, and the number of accessory genes varies among these SECs: TGEV has three accessory genes (non-structure 3a (NS3a), NS3b, and NS7); PEDV has one (ORF3); PDCoV has three (NS6, NS7, and NS7a); and SADS-CoV has three (NS3, NS7a, and NS7b) (Figure 1).

### 1.2. Pathogenicity of SECs in China

All four SECs mainly infect the small intestinal villous enterocytes of pigs and cause indistinguishable clinical signs, including diarrhea, vomiting, loss of appetite, dehydration, etc. [18,19,20,21,22,23,24]. They are deadly to neonatal piglets, while the disease severity decreases significantly in older and adult pigs. Generally, TGEV and PEDV are considered more virulent than PDCoV and SADS-CoV, although the pathogenicity of SADS-CoV in pigs is still controversial [14,23,25,26]. TGEV and PEDV cause severe clinical signs in nursing piglets under 2 weeks of age, with a mortality of up to 100%. The clinical signs in weaned to finisher pigs and pregnant sows are much milder, and the mortality rates are low [19,20,27,28,29,30,31,32]. PDCoV (HKU15 strain) was initially detected in a clinically normal pig in a molecular surveillance study in Hong Kong in 2009 [33]. The pathogenicity of PDCoV was first recognized in 2014 when it was found to be the cause of pig diarrheic outbreaks, initially on several farms in Ohio then throughout the United States (US) and globally [4,34,35]. Despite the prevalence and clinical severity of PDCoV are lower than those of PEDV, PDCoV experimentally infected other species, including chickens, turkeys, cattle, and mice [21,33,36,37,38,39]. In addition, PDCoV infected Haitian children and became a potential zoonotic pathogen [40]. SADS-CoV is a newly emerging, possibly bat-origin, swine enteric coronavirus discovered in 2016–2017 when large-scale pig diarrheic outbreaks occurred in Guangdong, China [23,26]. Although different SADS-CoV isolates were initially from the same region (Guangdong Province, China), the pathogenicity in piglets is still disputed. Huang’s group conducted several animal challenge experiments with SADS-CoV isolate CH/GD-01/2017 and found that the virus caused mild–moderate diarrhea or subclinical infections [14,23]. Two other groups have conducted challenge experiments with SADS-CoV isolates CN/GWDT/2017 and CH/GDS04, respectively, and both reported that the virus isolates caused watery diarrhea with mortality rates up to 100% in piglets [25,26]. We compared the genomic sequences of CH/GD-01/2017, CH/GDS04, and CN/GDWT/2017. The CH/GD-01/2017 shares 99.74% and 99.78% nucleotide (nt) identity with CH/GD04 and CN/GWDT/2017, respectively. Additionally, CH/GDS04 has a 3 nt deletion (4,561TTG4,563) in the ORF1a and a 3 nt insertion (24,788GTA24,790) in the M gene. These mutations may contribute to the pathogenicity discrepancy. Further studies are still needed to clarify whether the differences in the animal source, animal age, viral genomic sequence, or the combination of the above factors influence the SADS-CoV pathogenicity. The pathogenicity of SECs has been comprehensively reviewed previously [14,21,31,41].

### 1.3. Epidemiology of SECs in China

TGEV was first reported in the US in 1946 [42], and the disease with the characteristic features of transmissible gastroenteritis (TGE) was reported in China in the late 1960s [43]. A recent surveillance study reported that 21 (0.7%) of 2987 porcine diarrhea samples collected from five provinces in China from 2012 to 2018 were TGEV-positive by reverse transcription-PCR (RT-PCR) [44]. A possible reason for the low prevalence of TGEV is the emergence and widespread of porcine respiratory coronavirus (PRCV), a spike deletion mutant of TGEV, which is considered low pathogenic, and an imperfect natural vaccine for TGEV [45,46]. Although the positive rates of TGEV in diarrheic pigs were very low, there were still sporadic TGE outbreaks in Heilongjiang, Anhui, and Jiangsu provinces of China during 2012–2016 [28,29,30,47]. The geographic distribution of TGEV in the diarrheic samples from published data (from 2012 to 2018) [28,29,30,44,47] was shown in Figure 2A, while the true extent of the spread may not be fully reflected in the reported cases.

Since 1973, there have been continuous reports about TGE-like diarrhea endemics in which TGEV was not detected in China. Until 1984, PEDV was first identified as the etiogen for such outbreaks by fluorescence-labeled antibody and virus neutralization tests [48]. Before 2010, PEDV infections in China caused sporadic and endemic outbreaks. Since October 2010, highly virulent strains of PEDV emerged in Southern China, caused large PED outbreaks, and then spread rapidly throughout the country [49,50]. Currently, highly virulent PEDV strains in the genotype II group (G2) are dominant, whereas classical PEDV strains in the genotype I group (G1) are rare [44,51]. The genetic differences between the G1 and G2 PEDV strains, especially in the S protein, that is, the host receptor-binding protein and the most important viral protein for inducing neutralizing antibodies, account for the poor protection in pigs vaccinated with G1 PEDV-based vaccines (e.g., CV777) [52]. A recent PEDV surveillance study initiated by the New Hope Liuhe Co., Ltd. (Dezhou, China), one of the largest pig breeding enterprises in China, reported that 52.15% (158/303) of the farms were positive for PEDV with an overall detection rate of 63.95% (564/882) of the samples collected from 2017 to 2021 in sixteen provinces of China [53]. The phylogenetic analysis showed that the G2c subgroup was predominant, and the newly defined G2d strains were detected in the Sichuan, Hebei, and Henan provinces of China. Currently, PED is still the most devastating enteric disease in the pig industry of China due to its high mortality rate in nursing piglets. The geographic distribution of PEDV in China based on the data from Li et al., 2022 [53] is shown in Figure 2B, while the true extent of the spread may not be fully reflected in the reported cases.

In China, PDCoV was first reported in 2015; however, retrospective studies have shown that PDCoV emerged as a swine pathogen as early as 2004 [54,55,56,57]. A recent meta-analysis on the prevalence of PDCoV in China from 2015 to 2021 reported that overall PDCoV-positive rates were 13.61% (3828/25,977) [58]. Although PDCoV-positive rates were low, strains collected in China exhibited the broadest genetic diversity. Recently, we performed a phylogenetic analysis of 122 complete genomic sequences of PDCoV and found that all known PDCoV lineages, including US lineage, early Chinese lineage, Chinese lineage, and Vietnam/Laos/Thailand lineage, co-circulate in China [4]. Frequent recombination events between different lineages of PDCoV strains in China may complicate PDCoV epidemiology and lead to the emergence of PDCoV strains with changed pathogenicity and host tropism. For example, a highly virulent strain GNU-2105 (GenBank accession: OQ566226.1) that infected the entire intestines (from duodenum to colon) was reported in South Korea [59]. The geographic distribution of PDCoV in China based on the data from Wang et al. 2023 [58] was shown in Figure 2C, while the true extent of the spread may not be fully reflected in the reported cases.

SADS-CoV was first identified as the causative agent of severe diarrhea in suckling piglets in Guangdong province from 2016 to 2017 [23,25,26]. In 2018, a SADS-CoV strain CH/FJWT/2018 (GenBank accession: MH615810.1) was identified from fecal and small intestinal samples in Fujian province, China [60]. Subsequently, the porcine diarrhea associated viruses’ surveillance study [44] showed that SADS-CoV emerged in Fujian province as early as 2017. In February 2019, SADS-CoV re-emerged in pig farms in Guangdong province and claimed about 2000 deaths of pigs [61]. The strain CN/GDLX/2019 (GenBank accession: MK651076.1) was detected from pig intestinal samples and its S gene shared a higher nt identity (99.2 to 99.9%) with the previously reported SADS-CoV strains from Guangdong province than CH/FJWT/2018 (97.5%) from Fujian province [14], suggesting potential different origin of the SADS-CoVs in the two provinces. In May 2021, SADS-CoV emerged in pig herds in Guangxi province and killed more than 3000 piglets [62]. A new strain, SADS-CoV/Guangxi/2021 (GenBank Accession: ON911569.1), was detected from mixed samples. Phylogenetic analysis showed that SADS-CoV/Guangxi/2021 grouped with viruses from the Guangdong province, suggesting the ongoing transmission and evolution of the original Guangdong strains. To date, SADS-CoV has been reported in three provinces (Guangdong, Fujian, and Guangxi) in China. Two studies reported positive SADS-CoV rates in diarrheic samples collected from August 2016 to May 2017 in Guangdong province and in 2017 in Fujian province, respectively [44,63] (Figure 2D). The initial emergence and spread of SECs in China have been summarized in Table 1.

Circulating SECs in China are genetically diverse and widely distributed in the top provinces for pork production, except for SADS-CoV. Safe and effective vaccines against SECs are urgently needed to not only prevent SEC diseases and death in pigs but also reduce the potential risk to public health. The newly reported CCoV-HuPn-2018 and HuCCoV_Z19 suggest that TGEV may also have zoonotic potential [64]. SADS-CoV can infect primary human respiratory cells, suggesting its potential for zoonosis [65].

**Table 1 vaccines-12-00011-t001:** Initial emergence and spread of SECs in China.

Virus	First Year and Place of Emergence	Spread	References
TGEV	1956, Guangdong	Sporadic	[43]
PEDV (classical or G1)	1973, Shanghai	Sporadic	[66,67]
PEDV (highly virulent or G2)	2010, Guangdong	Epidemic	[49,50]
PDCoV	2004, Anhui	Sporadic	[54]
SADS-CoV	2016, Guangdong	Sporadic	[23]

## 2. SEC Vaccine Development in China

The most efficient vaccines against SECs are those that stimulate mucosal immunity in the intestine [11,68]. Neonatal piglets do not have time to develop immunity before they encounter deadly SECs. Therefore, immunization of sows with wild-type viruses or live attenuated vaccines (LAVs) has been widely used to provide neonatal piglets passive protection via colostrum and milk [69,70,71,72]. These LAVs provided higher than 90% litter protection in vaccinated sows. In general, there are several types of SEC vaccines: (1) inactivated vaccines, (2) LAVs generated by serial passage in cell cultures, (3) engineered LAVs, (4) subunit vaccines, (5) virus-like particles (VLPs), (6) nanoparticle trapped vaccines, and (7) virus-loaded microspheres.

### 2.1. Inactivated Vaccines

As a classical vaccine development route, inactivated vaccines to SECs are safe, and they have been widely used in humans and animals [73,74]. Although multiple inactivated vaccines against PEDV and TGEV are available in China (Table 2), they are only partially effective when they are used for sole vaccination of naïve sows/gilts. Their protection efficiency increases when they are used for priming and boosting sows/gilts prior to farrowing [75]. Researchers in China are focusing on the selection of new adjuvants to achieve excellent stimulation of the immune system. For instance, scientists found that inactivated TGEV (HN-2012, GenBank accession: OP434397.1) vaccine with mesoporous silica nanoparticles as adjuvant induced high TGEV-specific IgG and IgA antibody titers in mice and promoted the activation and maturation of porcine dendritic cells [76,77]. Xu et al. evaluated an inactivated PEDV (AH2012/12, GenBank accession: KU646831.1) vaccine adjuvanted with flagellin [78]. Sows immunized intranasally with this vaccine generated high levels of IgG, IgA, and viral neutralizing (VN) antibodies in the serum and colostrum, and their newborn piglets were protected from diarrhea post-PEDV challenge [78]. Similarly, sows immunized with the inactivated PDCoV NH strain (GenBank accession: KU981059.1) adjuvanted with aluminum hydroxide at the Houhai acupoint (a concave part between the anus and tail) generated high levels and long-lasting S protein-specific IgG and VN antibodies in serum [79]. The S protein-specific secretory IgA (sIgA) and IgG antibodies were also detected in the colostrum and milk of the immunized sows but lasted for a short period of time (~5 days). The IgG and VN antibodies were also detected from their piglets’ sera. Challenge experiment results showed that the inactivated PDCoV vaccine exhibited 87.1% protective rates against diarrhea in the piglets. Four other adjuvants, ODN2395 (oligodeoxynucleotides 2395), Imject^TM^ Alum Adjuvant, 206 adjuvant, and GSLS-NPs (ginseng stem–leaf saponins poly lactic-co-glycolic acid nanoparticles) were also evaluated with two inactivated PDCoV vaccine candidates HNZK-02 (GenBank accession: MH708123.1) and CH/XJYN/2016 (GenBank accession: MN064712.1), and one inactivated PEDV vaccine candidate ZJ-ZX2018-C10 (GenBank accession: MK250953.1) immunized in mice or pigs [80,81,82]. These inactivated vaccines induced high levels of PDCoV- or PEDV-specific IgG and VN antibody responses in mice or piglets. In addition, a biologically active adjuvant, *Bacillus subtilis* spores, was orally co-immunized twice with an inactivated PEDV strain (Zhejiang08) in 5-day-old piglets [83]. PEDV-specific IgA, IgG, and VN antibody levels were significantly induced in serum and mucosa, and the numbers of CD4^+^CD8^+^ memory T cells in the intestinal mucosal-associated lymphocytes also increased.

### 2.2. Live-Attenuated Vaccines (LAVs)

Inactivated vaccines typically induce a short-term neutralizing antibody response with comparatively low titers in colostrum and milk. On the contrary, LAVs can replicate in the enteric mucosa and generate a large number of viral particles to stimulate both mucosal and systemic immune responses to generate high levels of neutralizing antibodies in colostrum and milk [11]. Furthermore, LAVs can induce long-lasting and broad immunity to heterologous virus strains because they produce a full range of natural viral antigens and present the antigens to the immune system in the same way as natural infections. The conventional way to generate SEC LAV candidates is to use the cell culture-attenuation method. To date, five and two cell culture-attenuated PEDV and SADS-CoV strains, respectively, have been reported as LAV candidates in China. PEDV LAV candidates include FJzz1 (GenBank accession: MK288006.1) [84], SDSX16 (GenBank accession: MH117940.1) [85], CT (GenBank accession: MN114121.1) [86], YN (GenBank accession: KT021227.1) [87], and Zhejiang08 [88]. SADS-CoV LAV candidates include GDS04 [89] and CN/GDWT/2017 [90]. The degree of attenuation and immunogenicity of these LAV candidates were evaluated in neonatal or weaned pigs. However, all these studies lacked virulent virus challenge. LAV candidates FJzz1-F200 (the 200th passage (P200)), SDSX16-P75 (P75), CT-P120 (P120), GDS04-P100 (P100), and GDWT-P83 (P83) were all Vero cell-attenuated LAV candidates. The oral inoculation of suckling piglets resulted in lower fecal viral shedding and viral loads in the intestinal tissues and no obvious histopathological lesions [84,85,86,89,90]. The potential protective efficacy of FJzz1-F200 and SDSX16-P75 were instead conjectured from the induction of cytokines in porcine intestinal epithelial cells and serum IgG and IgA antibody responses in vaccinated neonatal piglets [84,85].

Reverse genetic technology is an ideal tool for the development of rationally designed LAVs. To date, several PEDV and PDCoV reverse genetic systems have been developed in China, including LAV candidates rAH2012/12 (the infectious clone-derived PEDV strain AH2012/12, GenBank accession: KC210145.1) [91], rAJ1102 (the infectious clone-derived PEDV strain AJ1102, GenBank accession: JX188454.1) [92], rCHM2013 (the infectious clone-derived PEDV strain CHM2013, GenBank accession: KM887144.1) [93], rBJ2011C (the infectious clone-derived PEDV strain BJ2011C, GenBank accession: KX066126.1) [93], and rCHN-HN-1601 (the infectious clone-derived PDCoV strain CHN-HN-1601) [94]. Two studies generated infectious clones with amino acid substitutions in the S protein of PEDV [95] or NS6 deletion of PDCoV [96], and led to enhanced virus replication efficiency in Vero cells or reduced virulence, respectively. Li et al. generated a PEDV-rotavirus chimeric virus with the genomic backbones of YN150 (GenBank accession: MZ581326.1) [97], in which the PEDV ORF3 was replaced by the rotavirus VP7 [97]. The immunogenicity and safety of the chimeric virus was evaluated in 10-day-old piglets [97]. Vaccinated piglets did not display any clinical signs but produced high levels of specific IgA, IgG, and VN antibodies against PEDV and rotavirus simultaneously.

A big challenge of applying LAVs in the field is the safety concern because an LAV can accumulate mutations and/or recombine with wild-type virus strains to generate new virulent variants, resulting in vaccination failure and disturbing surveillance. Hou et al. summarized the strategy for the rational design of PEDV LAV candidates using reverse genetics technology [98]. In general, mutant genes should not be essential for viral replication and immunogenicity; introducing attenuating mutations into multiple genes may make it hard for the virus to revert to virulent phenotypes. To avoid recombination, the Wang Lab recently generated a recombination-resistant PEDV mutant RMT by remodeling the transcriptional regulatory sequences-core sequences (TRS-CSs) [99]. This RMT can be further optimized and used as a platform to design recombination-resistant PEDV LAV candidates.

### 2.3. Subunit Vaccines

Compared with the whole virus vaccine approach, a subunit vaccine will only include specific antigens originating from disease-causing pathogens. For SECs, these antigens could be partial or full-length of S protein expressed in prokaryotic or eukaryotic cells [100]. Subunit vaccines possess several advantages, including high safety, no infectious viral nucleic acids, and providing a uniform antigen of a well-defined nature. However, due to low immunogenicity, it requires adjuvant or fusion with an immune enhancer [101,102]. To date, researchers have expressed partial S protein, including the collagenase equivalent domain (COE), receptor binding domain (RBD) and S1, or the full-length S protein of individual SECs using *E. coli*, *Lactobacillus*, *Bacillus subtilis*, yeast, 293T cells, virus expression system (e.g., vesicular stomatitis virus, swinepox virus, pseudorabies virus, baculovirus, and human adenovirus), or transgenic plant expression systems (Table 3). Among them, the immunogenicity of 14 vaccine candidates was evaluated in mice [103,104,105,106,107,108,109,110,111,112,113,114,115,116]. All these vaccine candidates induced high levels of the virus-specific IgG and VN antibodies in serum and cytokines in splenocytes. While only one candidate was identified to provide effective protection against the disease, post-homologous virulent virus challenge in mice [110]. The differences in the organization, distribution, and function of immune cells in intestinal effector sites (lamina propria) between pigs and mice suggest that the positive results from vaccine studies in mice should be confirmed in pigs [11]. Eight vaccine candidates were tested in pigs and provided effective protection against the disease post-homologous virulent virus challenge [117,118,119,120,121,122,123,124]. Table 3 summarizes the subunit vaccine candidates based on the SEC S proteins in China in recent years. Clearly, the identification and characterization of new adjuvants and the neutralizing epitopes to induce robust and long-lasting immunity are needed for subunit vaccine development.

### 2.4. Other Strategies

Generally, inactivated and subunit vaccines are safe, but they cannot induce long-term and broad protective immunity. LAVs can induce robust mucosal and systemic immune responses, but their safety is a concern. Therefore, other strategies, such as virus-like-particles (VLPs), nanoparticle-trapped vaccines, or virus-loaded microspheres against SECs, have drawn increasing attention in China in recent years. VLPs are structurally like native virions, containing one or more viral structural proteins but not viral nucleic acids. So, they have no risk of reversion to virulence and can elicit similar levels of robust humoral and cellular immune responses as LAVs [130,131,132]. VLP-based vaccines composed of the S, M, and E proteins against PEDV or PDCoV have been developed using a baculovirus expression vector system [133,134]. The immunogenicity of VLPs was evaluated in mice. Both VLPs could induce high levels of specific IgG and VN antibodies in serum and cytokines in splenocytes [133,134]. In addition, Guo et al. engineered a chimeric VLP vaccine candidate using the phage Qbeta coat protein presenting the universal epitope of coronavirus (conservative residues -WDYPKCDRA- located in nsp12), including PEDV and PDCoV [135]. The chimeric VLPs induced VN antibody responses against PEDV and PDCoV in mice. However, the immunogenicity and protective efficacy of these VLPs have not been tested in pigs.

Nanoparticles or microspheres are new vaccine delivery systems [136]. Furthermore, nanoparticles or microspheres can protect the entrapped or attached viral antigens from proteases-mediated degradation at mucosal surfaces and induce robust mucosal immune response [137,138,139]. Li et al. generated a nanoparticle-entrapped PEDV inactivated vaccine (strain AH2012/12) by using biodegradable poly (D,L-lactide-co-glycolide) and evaluated the immune responses in late-term pregnant sows and offspring piglets [140]. Serum levels of IgG, IgA, and VN antibodies were dramatically elicited in intranasally immunized sows and suckling piglets. More importantly, this vaccine provided an 80% protection rate against homologous virulent virus challenge in piglets. Additionally, Wen et al. have generated PEDV-loaded microspheres by using sucrose microspheres and centrifugal granulation technology and evaluated the immunogenicity in 4-week-old conventional weaned pigs [141]. IgG and VN antibodies in serum and saliva were significantly induced in the orally immunized pigs. No virus challenge study was performed in this study. Table 4 summarizes the VLPs and nanoparticles/microspheres vaccine delivery systems against SECs in China in recent years.

## 3. Status of Approved SEC Vaccines in China

Veterinary biological products in China are regulated by the Ministry of Agricultural and Rural Affairs. All animal vaccines used in China are licensed or permitted under the conventional product licensing requirement “veterinary drug production quality regulations (revised in 2020)” for assuring purity, potency, safety, and efficacy (https://www.gov.cn/gongbao/content/2020/content_5522524.htm; accessed on 25 October 2023). To date, nine vaccines against PEDV and TGEV have been approved, according to the National Veterinary Drug Database of China (http://vdts.ivdc.org.cn:8081/cx/; accessed on 25 October 2023). All the commercialized inactivated vaccines and LAVs for TGEV and PEDV were generated by serially passing the viruses in cell culture (Table 2). In addition, one PEDV subunit vaccine and three PDCoV inactivated vaccines are under clinical trials (Table 5). Nevertheless, new vaccine approvals will take several years, according to the licensing processes.

## 4. Conclusions and Future Perspectives

Currently, SECs still cause huge economic losses in the pork industry in China. In addition to good production management and strict biosecurity measures, the most effective way to prevent and control SECs is vaccination. Although many studies have been conducted on vaccine development and nine commercialized vaccines against TGEV and/or PEDV are available, PEDV remains the most prevalent enteric pathogen of pigs in China. There are several possible reasons: (1) The currently available vaccines (G1 genogroup) failed to provide full protection against the circulating strains (G2 genogroup); (2) Immunosuppressive diseases, such as porcine reproductive and respiratory syndrome and porcine circovirus-associated disease, active herds may drastically affect vaccine immunogenicity even with intense vaccination programs; (3) Antigenicity alteration of new emerging variants due to intense vaccination programs. Consequently, new vaccines against PEDV, such as recombination-resistant LAVs, are highly desirable. In addition, further studies are needed to elucidate the optimum immunization frequency, dose, route, and stage of pregnancy or lactation to elicit enough protective lactogenic immunity. To date, there is no approved vaccine against PDCoV or SADS-CoV, although PDCoV infection has been reported in all swine-producing areas in China. Intentional infection of pregnant sows/gilts with wild-type SECs to induce herd immunity may be an effective short-term solution. However, it may result in unknown outcomes, such as the spread of other swine pathogens on farms. Finally, further studies on SECs evolution, epidemiology, pathogenesis, and heterologous immunity are a prerequisite for the development of safe and effective vaccines to prevent and control SECs in China and other countries.

## Figures and Tables

**Figure 1 vaccines-12-00011-f001:**
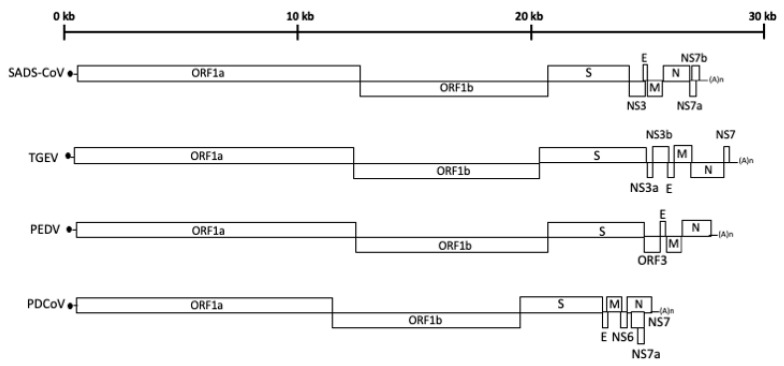
Schematic diagram showing the genomic organization of SECs: SADS-CoV, TGEV, PEDV, and PDCoV.

**Figure 2 vaccines-12-00011-f002:**
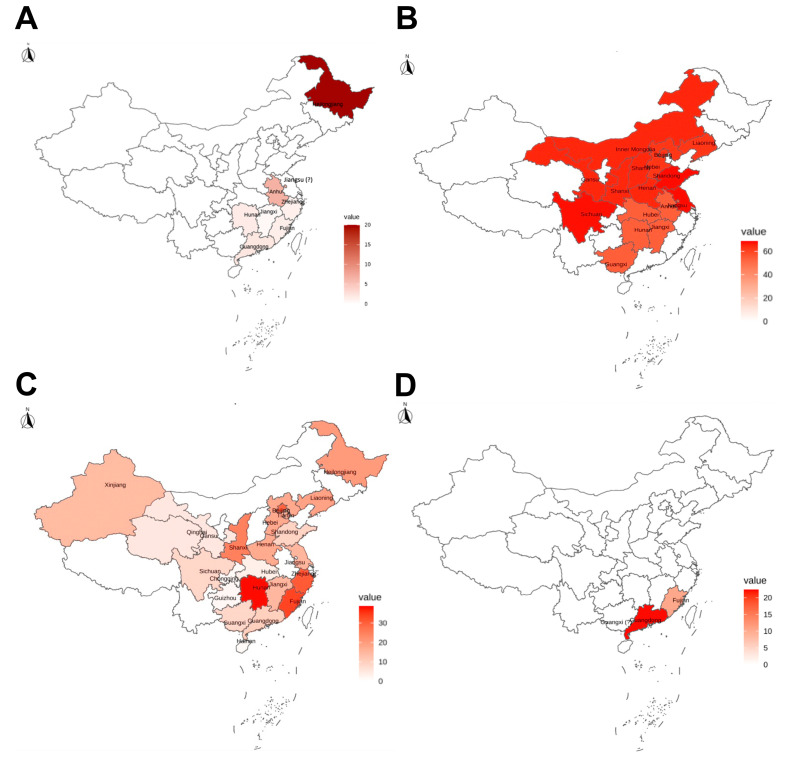
The geographic distribution of TGEV, PEDV, PDCoV, and SADS-CoV in the diarrheic samples in China. (**A**) The color of the map represents the positive rates of TGEV in total samples collected from 2012 to 2016. “Jiangsu (?)” means TGEV was successfully isolated from field samples in Jiangsu province, but no positive rates were reported. (**B**) The color of the map represents the positive rates of PEDV in total samples collected from 2017 to 2021. (**C**) The color of the map represents the positive rates of PDCoV in total samples collected from 2015 to 2021. (**D**) The color of the map represents the positive rates of SADS-CoV in total samples collected from 2016 to 2017. “Guangxi (?)” means SADS-CoV strain was identified from field samples in Guangxi province, but no positive rates were reported.

**Table 2 vaccines-12-00011-t002:** Commercialized SEC vaccines in China.

Vaccine Type	Strains	Route	Approval Number	Date of Approval
Inactivated vaccines of PEDV	CHYJ	Oral and nasal spray	(2023) New veterinary drug No. 02	12 January 2023
	XJ-DB2	IM	(2021) New veterinary drug No. 11	18 March 2021
Inactivated vaccines of TGEV and PEDV	WH-1 and AJ1102	IM	(2016) New veterinary drug No. 66	25 October 2016
	Hua and CV777	Houhai acupoint injection	(1999) New veterinary drug No. 45	-
Live vaccines of TGEV and PEDV	SD/L and LW/L	IM	(2018) New veterinary drug No. 37	3 July 2018
	SCJY-1 and SCSZ-1	IM and Houhai acupoint injection	(2017) New veterinary drug No. 64	27 December 2017
	WH-1R and AJ1102-R	IM	(2017) New veterinary drug No. 63	27 December 2017
	HB08 and ZJ08	IM and Houhai acupoint injection	(2015) New veterinary drug No. 57	18 November 2015
Live triple vaccines of TGEV, PEDV, and PRoV	Hua, CV777 and NX	Houhai acupoint injection	(2014) New veterinary drug No. 54	26 December 2014

Abbreviations in the table. TGEV: transmissible gastroenteritis virus; PEDV: porcine epidemic diarrhea virus; PRoV: porcine rotavirus; IM: intramuscularly; No.: number; -: no information on the official website. Note: all vaccine types, strains, approval numbers, and approval dates were retrieved from the national veterinary drug database of China (http://vdts.ivdc.org.cn:8081/cx/#) on 25 October 2023.

**Table 3 vaccines-12-00011-t003:** Subunit vaccine candidates based on the S protein of SECs in China.

Virus	Expression System	Antigen	Adjuvant or Immune Enhancer	Animal	Route	Amount of Challenge (Homologous Viruses)	Detection Indicators	References
TGEV	*L. plantarum* (NC8)	TGEV strain JS2012 (GenBank accession: KT696544.1) S protein	Dendritic cell-targeting peptide (FYPSYHSTPQRP)	Specific pathogen-free (SPF) mice	Oral	NR	(1) Percentage of B220^+^ IgA^+^ B cells in the small intestine lamina propria ↑ (2) Fecal sIgA ↑ (3) Serum IgG and VN antibodies ↑ (4) IFN-γ and IL-4 secretion in splenocytes ↑	[115]
	*L. plantarum* (NC8)	TGEV strain JS2012 S protein	Dendritic cell-targeting peptide (FYPSYHSTPQRP)	1-month-old crossbreed Junmu1 white piglets	Oral	NR	(1) Percentage of MHC-II^+^CD80^+^ B cells in the spleens and Peyer’s patches ↑ (2) Number of IgA^+^ B cells in the small intestine lamina propria ↑ (3) Percentage of CD3^+^CD4^+^ T cells in the small intestine lamina propria and mesenteric lymph nodes ↑ (4) Fecal sIgA ↑ (5) Serum IgG ↑ (6) Serum IL-4 and IL-17 ↑ (7) IL-4, IL-17, IFN-γ, and TGF-β in splenic lymphocytes, mesenteric lymph node lymphocytes, and ileum lamina propria lymphocytes ↑	[125]
	Swinepox virus-porcine kidney cells (PK-15, ATCC-CCL-33)	TGEV strain SHXB (GenBank accession: KP202848.1) S protein	NR	6-week-old BALB/C mice, 1-month-old piglets (Large White), and 1-day-old piglets	Oral, nasal, and IP	1 × 10^8^ PFU	(1) Mice: serum IgG, VN antibodies, IL-4, and IFN-γ ↑ (2) Pigs: serum IgG and VN antibodies ↑ (3) Piglets: mortality ↓ (4) Piglets: intestinal lesions ↓	[121]
	*B. subtilis* (WB800)	TGEV strain SHXB S protein	Spore coat protein	1-month-old Yorkshire piglets	Oral	NR	(1) Serum IgG and VN antibodies ↑ (2) Fecal sIgA ↑ (3) Activation of bone marrow dendritic cells ↑ (4) Numbers of ileum local lymphocytes ↑	[126]
	*L. acidophilus*	TGEV stain HN2002 (GenBank accession: AY587882.1) partial S protein (A and D antigenic sites)	NR	6-week-old female BALB/c mice	Oral	NR	Serum IgG, sIgA, IL-4, and IFN-γ ↑	[114]
PEDV	*E. coli* BL21 (DE3)	PEDV strain GDS01 (GenBank accession: KM089829.1) COE	Flagellin	4-week-old crossbred growing piglets (Duroc × Landrace × Big White)	Houhai acupoint	4 × 10^6^ PFU	(1) Serum IgG, IgA, VN antibodies, IFN-λ, and IL-4 ↑ (2) Saliva and face sIgA ↑ (3) Fecal viral load ↓	[119]
	*E. coli* BL21 (DE3)	PEDV strain CH2019 partial S protein (499–789aa + 1368–1378aa)	Pseudomonas aeruginosa exotoxin A without domain III	6-week-old female BALB/c mice	IN	NR	(1) Serum IgG, IgA, and VN antibodies ↑ (2) Mucosal IgA in the intestinal contents ↑ (3) Spleen lymphocyte proliferation ↑ (4) IFN-λ, IL-2, IL-4, and IL-10 secretion in splenocytes ↑	[109]
	Expi293FTMcells and *E. coli* BL21 (DE3)	PEDV strain AH2012/12 COE	Layered double hydroxide	6-week-old BALB/c mice	SC	NR	(1) CD3^+^CD4^+^ T cells ↑ (2) CD3^+^CD8^+^ T cells ↑ (3) Spleen lymphocyte proliferation ↑ (4) IFN-γ and IL-4 secretion in splenocytes ↑	[108]
	Yeast (*P. pastoris* GS115)	PEDV strain CV777 S1	NR	5-week-old BALB/c mice and piglets	Oral	NR	(1) Mice: IgG in serum and sIgA in feces ↑ (2) Piglets: sIgA in feces ↑	[127]
	*L. plantarum* (NC8)	PEDV strain CH/ZY/11 (GenBank accession: JQ257007.1) S protein	Dendritic cell-targeting peptide (FYPSYHSTPQRP)	6-week-old SPF mice	Lavage	NR	(1) IgA^+^B220^+^ B cells ↑ (2) Fecal sIgA ↑ (3) Serum IgG ↑ (4) Serum and fecal VN antibodies ↑	[103]
	*L. casei 393*	PEDV strain LJB/15 COE	Dendritic cell-targeting peptide (FYPSYHSTPQRP)	6-week-old female SPF BALB/c mice	Oral	NR	(1) Serum IgG and VN antibodies ↑ (2) Fecal sIgA ↑ (3) Spleen lymphocyte proliferation ↑ (4) IFN-γ and IL-4 secretion in splenocytes ↑	[111]
	Swinepox virus-porcine kidney cells (PK-15, ATCC-CCL-33)	PEDV strain SQ2014 (GenBank accession: KP728470.1) partial S protein (386–815aa)	NR	1-month-old piglets (Large White)	SC and orally (1:1 volume ratio)	1 × 10^6^ PFU	(1) Serum IgG, IgA, and VN antibodies ↑ (2) IFN-γ and IL-4 in peripheral blood lymphocytes ↑ (3) Mortality of piglets ↓ (4) Intestinal lesions ↓	[122]
	*L. casei 393*	PEDV strain HLJ-2012 COE	Microfold cell–targeting peptide Co1	6-week-old female SPF BALB/c mice	Oral	NR	(1) Serum IgG and VN antibodies ↑ (2) Intestinal mucus and genital tract sIgA ↑ (3) IFN-γ and IL-4 secretion in splenocytes ↑ (4) Spleen lymphocyte proliferation ↑	[112]
	*L. casei 393*	PEDV strain LJB/15 COE	Dendritic cell-targeting peptide (FYPSYHSTPQRP)	Crossbred piglets (Large White)	Oral	5 × 10^6^ PFU	(1) Mucus extract sIgA and serum IgG ↑ (2) TLR-4, TLR-9, and TGF-β in mesenteric lymph nodes ↑ (3) Mortality of piglets ↓ (4) Intestinal lesions ↓ (5) Virus shedding ↓	[117]
	*L. casei 393*	PEDV strain HLJ-2012 COE	Microfold cell–targeting peptide Co1 and dendritic cell-targeting peptide (FYPSYHSTPQRP)	6-week-old female SPF BALB/c mice	Oral	NR	(1) Serum IgG and VN antibodies ↑ (2) Intestinal mucus, genital tract, and fecal sIgA ↑ (3) IFN-γ and IL-4 secretion in splenocytes ↑ (4) Spleen lymphocyte proliferation ↑	[105]
	Vesicular stomatitis virus-baby hamster kidney cells (BHK-21) (ATCC-CCL-10)	PEDV strain CHN/SHANGHAI/2012 (GenBank accession: MG83701.1) partial S protein (19 aa deletion in cytoplasmic tail)	MONTANIDE^TM^ IMS 1313 VG NPR adjuvant, MONTANIDE^TM^ ISA 206 VG adjuvant, or MONTANIDE^TM^ ISA15A VG adjuvant	4-week-old healthy Bama minipigs and sows	IM	1 × 10^2^ TCID_50_	(1) Piglets: serum IgG and VN antibodies ↑ (2) Sows: serum and colostrum VN antibodies ↑ (3) Mortality of piglets ↓ (4) Piglets: fecal score ↓ (5) Piglets: virus shedding ↓	[118]
	Ad5Max adenovirus vector system-HEK293A cells	PEDV strain CH/HBXT/2018 (GenBank accession: MH816969.1) S protein	NR	4-week-old piglets	IM	1 × 10^5^ TCID_50_	(1) Serum IgG, IgA, and VN antibodies ↑ (2) Fecal viral shedding ↓ (3) Fecal scores ↓	[120]
	*B. subtilis* (WB800)	PEDV strain Zhejiang08 COE	NR	SPF piglets (Duroc and Landrace and Yorkshire)	Oral	NR	(1) The area of Peyer’s patches ↑ (2) The villi length of ileum ↑ (3) Saliva and fecal sIgA ↑ (4) Serum IgG ↑ (5) CD3^+^ T lymphocytes and IgA^+^ cells in ileum ↑ (6) CD3^+^CD4^+^ T cells in small intestinal mucosa ↑ (7) IL-1β and IL-10 levels ↑ (8) Plaque reduction neutralization test ↑	[128]
	*L. acidophilus* (SW1)	PEDV strain CH-SDBZ-1-2015 (GenBank accession: KU133232.1) S1	NR	6-week-old female BALB/c mice or pregnant sows	Oral	NR	(1) Mice: serum IgG, sIgA, IFN-γ, and IL-4 ↑ (2) Sow: serum IgG ↑ (3) Sow: sIgA in colostrum ↑	[129]
	*L. johnsonii* (6332)	PEDV strain HLJ-2012 COE	NR	Large white sows at 90 days of gestation and 4-day-old piglets	Oral	1 × 10^4^ TCID_50_	(1) Monocyte-derived dendritic cell (MoDC) maturation ↑ (2) Sow: serum IgG, IgA, IgM, IL-2, IL-4, IL-6, IL-12, and IFN-γ ↑ (3) IgG, sIgA in colostrum ↑ (4) Piglets: serum IgG and sIgA ↑ (5) Piglets: nasal mucosa and rectal mucosa IgG and sIgA ↑ (6) Piglets: fecal viral shedding ↓ (7) Piglets: clinical signs of fecal consistency ↓ (8) Piglets: intestinal lesions ↓	[123]
	*L. casei* deficient in *upp* genotype *(*∆*upp ATCC 393)*	PEDV strain LJB2019 S1	NR	35-day-old female SPF BALB/c mice	Oral	NR	(1) Serum IgG and VN antibodies ↑ (2) Nasal, tears, genital tract, intestinal mucus, and fecal sIgA ↑ (3) Serum IL-2, IFN-γ, IL-4, IL-12, IL-10, and IL-17 ↑ (4) Spleen lymphocyte proliferation ↑	[113]
	Human adenovirus type 5 (Ad5) vector-HEK293 cells	PEDV strain CH/HBXT/2018 (GenBank accession: MH816969.1) S or S1 and PEDV strain CH/HNPJ/2017 (GenBank accession: MF152604.1) S or S1	NR	6–8 weeks-old SPF female BALB/c mice	IM	NR	(1) Serum IgG, IL-2, TNF-α, and VN antibodies ↑ (2) Percentages of splenic CD8^+^ T cells ↑	[107]
PDCoV	Human adenovirus type 5 (Ad5) vector-HEK293 cells	PDCoV strain CH/XJYN/2016 (GenBank accession: MN064712.1) S or S1	NR	6–8 weeks-old SPF female BALB/c mice	IM	NR	(1) Serum IgG, IL-2, TNF-α, and VN antibodies ↑ (2) Percentages of splenic CD8^+^ T cells ↑	[106]
	Baculovirus-High Five (Hi5) and *spodoptera frugiperda* (sf9) cells	PDCoV strain CZ2020 (GenBank accession: OK546242.1) S or RBD	M103 adjuvant or GEL 01 adjuvant	6-week-old female BALB/c mice, 3-day-old piglets, and sows	IM	1 × 10^7.75^ TCID_50_ or 1 × 10^7^ TCID_50_ or 2 × 10^5^ TCID_50_	(1) Mice: serum IgG, IFN-γ, IL-4, and VN antibodies ↑ (2) Percentages of mice splenocyte CD3^+^CD4^+^ T cells, CD3^+^CD8^+^ T cells, B220^+^CD19^+^ B lymphocytes, and CD3^−^CD49b^+^ NK cells ↑ (3) Piglets: serum IgG, sIgA, and VN antibodies ↑ (4) Piglets peripheral blood mononuclear cells (PBMCs) lymphocyte proliferation ↑ (5) Piglets: Il-4 and IFN-γ mRNA levels in PBMCs ↑ (6) Sow: milk IgG and VN antibodies ↑ (7) Piglets: virus shedding ↓ (8) Piglets: no pathological changes in intestinal tissues	[124]
	Pseudorabies virus-BHK-21 cells	PDCoV strain CH/Sichuan/S27/2012 (GenBank accession: KT266822.1) S	NR	6–8-week-old female BALB/c mice	IM	NR	(1) Serum IgG, VN antibodies, IL-4, and IFN-γ ↑ (2) Spleen lymphocyte proliferation ↑ (3) Percentage of CD4^+^ and CD8^+^ T cells ↑	[104]
	*L. lactis (NZ9000)*	PDCoV strain CH/XJYN/2016 (GenBank accession: MN064712.1) S1	Microfold cell–targeting peptide Co1	6-week-old SPF female BALB/c mice	Oral	NR	(1) Serum IgG, IgA, IFN-γ, IL-2, IL-4, and IL-17 ↑ (2) Fecal sIgA ↑ (3) Intestinal fluid VN antibodies ↑ (4) Percentage of CD3^+^, CD3^+^CD4^+^, and CD3^+^CD8^+^ T cells ↑ (5) Spleen lymphocyte proliferation ↑	[116]
	Baculovirus-sf9 cells	PDCoV strain HNZK-02 RBD	CPG2395 or aqueous adjuvant	6-week-old female BALB/c mice	SC	3 × 10^8^ TCID_50_	(1) Serum IgG, IgA, and VN antibodies ↑ (2) IFN-γ, IL-2, IL-4 secretion in splenocytes ↑ (3) Spleen lymphocyte proliferation ↑ (4) CD4^+^CD8^+^ T cells ↑ (5) Central memory T cells ↑ (6) Intestinal pathological lesions ↓	[110]

Abbreviations in the table. NR: not reported; VN: viral neutralizing; COE: collagenase equivalent domain; RBD: receptor binding domain; TLR: Toll-like receptor; TGF-β: transforming growth factor-β; aa: amino acid; IM: intramuscularly; IN: intranasally; IP: intraperitoneal; SC: subcutaneously; PFU: plaque formation unit; TCID_50_: 50% tissue culture infectious dose; *L. casei*: *Lactobacillus casei*; *L. johnsonii*: *Lactobacillus johnsonii*; *L. acidophilus*: *Lactobacillus acidophilus*; *L. lactis*: *Lactococcus lactis*; *B. subtilis*: *Bacillus subtilis*; *L. plantarum*: *Lactobacillus plantarum*; *P. pastoris*: *Pichia pastoris*; *E. coli*: *Escherichia coli*; ↑: up; ↓: down.

**Table 4 vaccines-12-00011-t004:** VLPs and nanoparticles/microspheres vaccine delivery systems.

Strategy	Virus	Expression System	Antigen	Adjuvant or Immune Enhancer	Animal	Route	Amount of Challenge (Homologous Viruses)	Detection Indicators	References
Virus-like-particles	PEDV	Baculovirus-Sf9 cells	S, M, and E proteins	NR	6-week-old female BALB/c mice	IM	NR	(1) Serum VN antibodies ↑ (2) IL-4 secretion in CD4^+^ T cells ↑	[134]
	PDCoV	Baculovirus-sf9 cells	S, M, and E proteins	Freund’s adjuvant	8-week-old female BALB/C mice	IM	NR	(1) Serum IgG and VN antibodies ↑ (2) IFN-γ and IL-4 secretion in splenocytes ↑	[133]
Nanoparticle trapped vaccines	PEDV	NR	Poly (D,L-lactide-co-glycolide) nanoparticle–entrapped inactivated PEDV strain (AH2012/12)	ISA201 adjuvant	Pregnant sows and piglets	IN	100 LD_50_	(1) Sows: serum IgG and VN antibodies ↑ (2) Sows: colostrum IgG, IgA and VN antibodies ↑ (3) Sows: lymphocyte proliferation, IFN-γ production, and mRNA levels in PBMCs ↑ (4) Piglets: serum IgG and VN antibodies ↑ (5) Piglets: fecal viral shedding ↓ (6) Piglets: clinical signs ↓ (7) Piglets: intestinal lesions ↓ (8) Piglets: mortality rates ↓	[140]
Virus-loaded microspheres	PEDV	NR	Freeze-dried powders containing PEDV attenuated strain were loaded on sucrose microspheres	NR	4-week-old weaned piglets	Oral	NR	(1) Serum IgG and VN antibodies ↑ (2) Saliva IgA ↑	[141]

Abbreviations in the table. NR: not reported; VN: viral neutralizing; IM: intramuscularly; IN: intranasally; LD_50_: 50% lethal dose; ↑: up; ↓: down.

**Table 5 vaccines-12-00011-t005:** SEC vaccine candidates in clinical trials in China.

Project Name	Strain	Applicant	Term of Validity
Genetically engineered subunit vaccine of PEDV	CHO-3G12	Hailong (Zhejiang) Biotechnology Co., Ltd. (Shaoxing, China), and Institute of animal health, Guangdong academy of agricultural sciences.	21 July 2023–20 July 2025
Inactivated vaccine of PEDV	ZJ/15	Zhejiang University, Jiangsu Academic Science of Agriculture, Meibaolong (Zhejiang) Biotechnology Co., Ltd. (Jinhua, China), and Hongsheng (Zhejiang) Biotechnology Co., Ltd. (Shaoxing, China)	21 April 2023–20 April 2025
Inactivated triple vaccine of TGEV, PEDV and PDCoV	SHXB, ZJ/15 and LYG/14	Zhejiang University, Jiangsu Academic Science of Agriculture, and Hongsheng (Zhejiang) Biotechnology Co., Ltd. (Shaoxing, China)	19 December 2022–18 December 2024
Inactivated vaccine of PDCoV	LYG/14	Jiangsu Academic Science of Agriculture, Zhejiang University, Hongsheng (Zhejiang) Biotechnology Co., Ltd. (Shaoxing, China), and Yisikang (Chengdu) Pharmaceutical Technology Co., Ltd. (Chengdu, China)	6 February 2023–5 February 2025
Inactivated vaccine of PEDV	HeN2017	Henan Animal Disease Control Center (Zhengzhou, Henan, China), Nuoweilihua (Shanxi) Biotechnology Co., Ltd. (Xianyang, China), Longxing (Yangling) Technology Development Co., Ltd. (Xianyang, China), and Tianhexin (Suzhou) Biotechnology Co., Ltd. (Suzhou, China)	2 March 2023–1 March 2025
Inactivated vaccine of PEDV and PRoV	HB17 and JS01	Keqian (Wuhan) Biology Co., Ltd. (Wuhan, China)	31 May 2022–30 May 2024
Inactivated vaccine of PDCoV and PEDV	AK-R and WN-R	Lufang (Yangling) Bioengineering Co., Ltd. (Xianyang, China), Nuoweilihua (Shaanxi) Biotechnology Co., Ltd. (Xianyang, China), Shengwei (Fujian) Biotechnology Co., Ltd. (Nanping, China), and Beijing University of Agriculture.	26 January 2022–25 January 2024

Note: data retrieved from the national veterinary drug database of China (http://vdts.ivdc.org.cn:8081/cx/) on 25 October 2023.

## Data Availability

The data presented in this study are openly available at GenBank (https://www.ncbi.nlm.nih.gov/nucleotide/, accessed on 24 October 2023) and the websites provided in this manuscript.

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
