# Peer review of "Prevention and Control of Swine Enteric Coronaviruses in China: A Review of Vaccine Development and Application"

_vaccines, 2023, doi:10.3390/vaccines12010011_

Round 1
Reviewer 1 Report
Comments and Suggestions for Authors
The article written by Kong and entitled "Prevention and control of swine enteric coronaviruses in China: A review of vaccine development and application" is a review relating to indirect prophylaxis against porcine enteric coronaviruses. The work is excellently written, although some sections could be improved with additional information. Below are my specific comments.
1) Line 23: Authors wrote “aiming to provide suggestions for better prevention and control of SECs in China”. Why couldn't all this information be useful for the control and prevention of these infections in other countries too?
2) Introduction: Further basic information would be necessary about the viruses in question, specifying the biological cycle, transmission, presence of the virus in wild animals. doi: 10.7589/JWD-D-21-00196
3) Line 47: Genomic organization: This section can also be improved by listing the main differences with other porcine coronaviruses (PRCV) and other species.
4) Line 358-360: As point 1.
Author Response
Please see the attachment.
- Summary
Thank you very much for taking the time to review this manuscript. Please find the detailed responses below and the corresponding revisions/corrections in track changes in the re-submitted files.
- Point-by-point response to Comments and Suggestions
Comments 1: Line 23: Authors wrote “aiming to provide suggestions for better prevention and control of SECs in China”. Why couldn't all this information be useful for the control and prevention of these infections in other countries too?
Response 1: Thanks for pointing this out. We agree with this comment. Therefore, we have added “and other countries” in line 23 of the revised manuscript.
Comments 2: Introduction: Further basic information would be necessary about the viruses in question, specifying the biological cycle, transmission, presence of the virus in wild animals. doi: 10.7589/JWD-D-21-00196
Response 2: Agree. We have, accordingly, added the information of viral biological cycle and the virus infection of wild animals in line 34-36 and in line 40-42. The transmission of the viruses was mentioned in line 32-34: “α-CoVs and β-CoVs infect mammals, while γ-CoVs and δ-CoVs primarily infect birds with some mammalian spillover [1-4]”.
Comments 3: Line 47: Genomic organization: This section can also be improved by listing the main differences with other porcine coronaviruses (PRCV) and other species.
Response 3: Thanks for your suggestions. The main differences between TGEV and PRCV have been mentioned in line 104-107.
Comments 4: Line 358-360: As point 1.
Response 4: Thanks for pointing this out. We agree with this comment. Therefore, we have added “in China and other countries” in line 409 of the revised manuscript.

Reviewer 2 Report
Comments and Suggestions for Authors
I reviewed the manuscript entitled “Prevention and control of swine enteric coronaviruses in China: A review of vaccine development and application”.
Overall, I consider that the concept of this review is interesting, considering the relevance of enteric coronaviruses. However, I consider that several issues should be properly addressed to improve the quality of this manuscript. These are some of my suggestions:
A) I consider that different viral diseases should be described in separated sections.
B) The addition of a phylogenetic tree describing the genetic variability of different viral diseases in China. In this sense, describe more details about the evolution of different genes. How it can impact the development of vaccine subunits? And the use of LAVs
C) In the description of different vaccine candidates, highlight the ones tested in swine versus the ones developed in other species like mice. Provide more details about the effectiveness of different vaccine candidates.
D) Genetically, in terms of the vaccine strains used in China, how different these are, compared with the current strains circulating in China? How intense vaccination programs are? Is it creating a selective pressure for the emergence of new variants? Please, discuss these possibilities.
Author Response
Please see the attachment.
- Summary
Thank you very much for taking the time to review this manuscript. Please find the detailed responses below and the corresponding revisions/corrections in track changes in the re-submitted files.
- Point-by-point response to Comments and Suggestions for Authors
Comments 1: I consider that different viral diseases should be described in separated sections.
Response 1: Thanks for your suggestions. Pathogenicity of these four swine enteric coronaviruses (SECs) have been described in section 1.2. These enteric coronaviruses cause similar clinical signs. Therefore, we do not have the description of different viral diseases in separated sections.
Comments 2: The addition of a phylogenetic tree describing the genetic variability of different viral diseases in China. In this sense, describe more details about the evolution of different genes. How it can impact the development of vaccine subunits? And the use of LAVs
Response 2: In this review, we are focusing on the development and application of vaccines against SECs in China. The genetic diversity and phylogenetic analysis of SECs have been comprehensively reviewed recently and cited in this review [1-4]. There is no doubt that genetic diversity and evolution are major stumbling blocks to the successful development of an effective vaccine. Therefore, to cope with genetic diversity and evolution, a recombination-resistant LAV platform was generated recently (described in line 286-292) and the S proteins of circulating strains were expressed as vaccine subunits (Table 3).
References:
- Wang PH, Nawal Bahoussi A, Tariq Shah P, Guo YY, Wu C, Xing L. Genetic comparison of transmissible gastroenteritis coronaviruses. Front Vet Sci. 2023 Apr 17;10:1146648. doi: 10.3389/fvets.2023.1146648.
- Li M, Pan Y, Xi Y, Wang M, Zeng Q. Insights and progress on epidemic characteristics, genotyping, and preventive measures of PEDV in China: A review. Microb Pathog. 2023 Aug;181:106185. doi: 10.1016/j.micpath.2023.106185.
- Kong F, Wang Q, Kenney SP, Jung K, Vlasova AN, Saif LJ. Porcine Deltacoronaviruses: Origin, Evolution, Cross-Species Transmission and Zoonotic Potential. Pathogens. 2022 Jan 9;11(1):79. doi: 10.3390/pathogens11010079.
- Yang YL, Yu JQ, Huang YW. Swine enteric alphacoronavirus (swine acute diarrhea syndrome coronavirus): An update three years after its discovery. Virus Res. 2020 Aug;285:198024. doi: 10.1016/j.virusres.2020.198024.
Comments 3: In the description of different vaccine candidates, highlight the ones tested in swine versus the ones developed in other species like mice. Provide more details about the effectiveness of different vaccine candidates.
Response 3: Agree. We have, accordingly, added more details about the effectiveness of different vaccine candidates tested in swine versus in mice in the revised manuscript (lines 304-311).
Comments 4: Genetically, in terms of the vaccine strains used in China, how different these are, compared with the current strains circulating in China? How intense vaccination programs are? Is it creating a selective pressure for the emergence of new variants? Please, discuss these possibilities.
Response 4: Agree. We have, accordingly, discussed the differences between vaccine strains and circulating strains, vaccination programs, and selective pressure for the emergence of new variants in the conclusion and future perspective section in the revised manuscript line 394-399.

Reviewer 3 Report
Comments and Suggestions for Authors
This is an ambitious plan to summarize the R and D activities of vaccines against swine enteric coronaviruses in China. Some of the studies using mice are also included.
In general, the writing and the overall structures of this manuscript is clear and aided with various tables. I would suggest the following minor revisions for better reference.
Section 1.2 Epidemiology of SEC in China. I suggest using a table to summarize the first year of emergence (including those from retrospective studies) for each virus in China, the place of first found, and the situation of spread from A to B or to the whole country (example how long it take, by what vehicle or means?). This is an essential compliment to Figure 1, which is presumedly the results of recent 10 year using molecular method.
line 171: confirm "Jiansu (?)".
Section 2.4: tabulate the results, like those in Table 2, for easier glance.
Author Response
Please see the attachment.
- Summary
Thank you very much for taking the time to review this manuscript. Please find the detailed responses below and the corresponding revisions/corrections in track changes in the re-submitted files.
- Point-by-point response to Comments and Suggestions
Comments 1: Section 1.2 Epidemiology of SEC in China. I suggest using a table to summarize the first year of emergence (including those from retrospective studies) for each virus in China, the place of first found, and the situation of spread from A to B or to the whole country (example how long it take, by what vehicle or means?). This is an essential compliment to Figure 1, which is presumedly the results of recent 10 year using molecular method.
Response 1: Agree. We have, accordingly, made a table (Table 1 in the revised manuscript) to summarize the basic information of SECs’ initial emergence and spread in China.
Comments 2: line 171: confirm "Jiansu (?)".
Response 2: Thanks for pointing this out. We have double checked the information published by Guo et al., 2020, and we confirmed that no positive rates were mentioned in the paper.
Reference: Guo R, Fan B, Chang X, Zhou J, Zhao Y, Shi D, Yu Z, He K, Li B. Characterization and evaluation of the pathogenicity of a natural recombinant transmissible gastroenteritis virus in China. Virology. 2020 Jun;545:24-32. doi: 10.1016/j.virol.2020.03.001.
Comments 3: Section 2.4: tabulate the results, like those in Table 2, for easier glance.
Response 3: Agree. We have, accordingly, made results in section 2.4 into a table (Table 4 in the revised manuscript).

Round 2
Reviewer 2 Report
Comments and Suggestions for Authors
I like to thank the authors for their responses. At this point, I don't have more concerns about this manuscript.